# A New Structural Model of Apolipoprotein B100 Based on Computational Modeling and Cross Linking

**DOI:** 10.3390/ijms231911480

**Published:** 2022-09-29

**Authors:** Kianoush Jeiran, Scott M. Gordon, Denis O. Sviridov, Angel M. Aponte, Amanda Haymond, Grzegorz Piszczek, Diego Lucero, Edward B. Neufeld, Iosif I. Vaisman, Lance Liotta, Ancha Baranova, Alan T. Remaley

**Affiliations:** 1Lipoprotein Metabolism Laboratory, National Heart, Lung, and Blood Institute, National Institutes of Health, Bethesda, MD 20892, USA; 2School of Systems Biology, George Mason University, Manassas, VA 20110, USA; 3Department of Physiology and Saha Cardiovascular Research Center, University of Kentucky, Lexington, KY 40536, USA; 4Proteomics Core Facility, National Heart, Lung, and Blood Institute, National Institutes of Health, Bethesda, MD 20892, USA; 5Center for Applied Proteomics and Molecular Medicine, George Mason University, Manassas, VA 20110, USA; 6Biophysics Core Facility, National Heart, Lung, and Blood Institute, National Institutes of Health, Bethesda, MD 20892, USA; 7Research Center for Medical Genetics, 115522 Moscow, Russia

**Keywords:** apolipoprotein B100, very low-density lipoprotein, LDL receptor ligand, lipovitellin, homology modeling, DSSO cross-linker, cardiovascular disease, ITASSER, divide and conquer algorithm

## Abstract

ApoB-100 is a member of a large lipid transfer protein superfamily and is one of the main apolipoproteins found on low-density lipoprotein (LDL) and very low-density lipoprotein (VLDL) particles. Despite its clinical significance for the development of cardiovascular disease, there is limited information on apoB-100 structure. We have developed a novel method based on the “divide and conquer” algorithm, using PSIPRED software, by dividing apoB-100 into five subunits and 11 domains. Models of each domain were prepared using I-TASSER, DEMO, RoseTTAFold, Phyre2, and MODELLER. Subsequently, we used disuccinimidyl sulfoxide (DSSO), a new mass spectrometry cleavable cross-linker, and the known position of disulfide bonds to experimentally validate each model. We obtained 65 unique DSSO cross-links, of which 87.5% were within a 26 Å threshold in the final model. We also evaluated the positions of cysteine residues involved in the eight known disulfide bonds in apoB-100, and each pair was measured within the expected 5.6 Å constraint. Finally, multiple domains were combined by applying constraints based on detected long-range DSSO cross-links to generate five subunits, which were subsequently merged to achieve an uninterrupted architecture for apoB-100 around a lipoprotein particle. Moreover, the dynamics of apoB-100 during particle size transitions was examined by comparing VLDL and LDL computational models and using experimental cross-linking data. In addition, the proposed model of receptor ligand binding of apoB-100 provides new insights into some of its functions.

## 1. Introduction

Lipoproteins are particles in the blood consisting of lipids and amphipathic proteins that carry various lipophilic components and are known to be causally linked to the development of atherosclerosis [1,2] and other pathophysiologic processes [3,4,5,6]. The outer layer of lipoproteins is formed from phospholipids, apolipoproteins, and unesterified cholesterol, and its hydrophobic core contains triglycerides (TG) and cholesterol esters (CE). Apolipoproteins are amphipathic proteins that stabilize lipoprotein particles and also serve as ligands for receptors and as modulators of lipid-modifying enzymes.

In humans, apolipoprotein B (apoB) is the major apolipoprotein on TG-rich lipoproteins (TRL). ApoB-100, the full length-form of the protein, is synthesized in hepatocytes and is initially secreted on very low-density lipoprotein (VLDL) but is retained on the particles after conversion of larger VLDL to smaller low-density lipoprotein (LDL) particles by lipolysis. ApoB-48 is a truncated form of the protein that is produced in the intestine and is found on chylomicrons. ApoB-100 is the largest of all the apolipoproteins with a molecular mass of 515 KDa and contains 4536 amino acids, while ApoB-48 only contains the first 2152 amino acids, which is approximately 48% of the N-terminal sequence [7,8,9]. Our understanding of the apoB-100 secondary and tertiary structure is limited because of its large size and likely dynamic structure.

ApoB-100 is a known member of a large lipid transfer protein superfamily, which includes the microsomal triglyceride transfer protein (MTP), lipovitellin, and several apolipophorins [10,11]. The crystal structure of silver lamprey lipovitellin and MTP resembles a pyramidal funnel [12,13,14,15]. The N-terminal domain of lipovitellin forms a half β-barrel shape at the pyramid’s apex. Two antiparallel β-sheets are lined on two sides of the pyramid with a cavity in the center, and another antiparallel β-sheet covers the open base of the cavity. Two curved layers of α-helices support the structure by surrounding the sides [14]. The first homology models of apoB-100 and MTP were made according to lipovitellin’s structure and covered the first 1000 residues of apoB [16,17,18]. The latest theoretical model of apoB-100 was proposed in 2006. This model divided the total protein into eight domains, and a 3D structure was proposed for each part [19]. Despite protein data bank expansion, new modeling algorithms, and experimental methods, there is no complete and accurate model for the entire apoB-100 protein.

CryoEM has recently been used to investigate lipoproteins’ structure and showed that lipoprotein particles substantially vary in shape. Large particles like VLDL tend to be more spherical, the smaller intermediate-density lipoprotein (IDL) particles are polyhedral, and electron micrographs of LDL reveal spherical, ellipsoidal, and discoidal particles [20,21,22,23]. The discoidal LDL particles show a low-density lipid in the core encircled by high-density apolipoproteins at the outer surface [20,21].

Given its great medical relevance for cardiovascular disease, we describe a new structural model of apoB-100 based on the latest computational modeling and new cross-linking data. Our approach is based on the “divide and conquer” algorithm, in which we first divided the apoB-100 sequence into smaller subunits and domains using PSIPRED [24,25]. Several computational models were then made for each subunit and domain by applying homology modeling and threading. Selected models were refined, energy minimized, and finally validated based on cross-links obtained using disuccinimidyl sulfoxide (DSSO), a mass spectrometry cleavable cross-linker, and the known disulfide bonds in apoB-100.

## 2. Results and Discussion

As described below and in the method section, apoB-100 was divided into five major subunits and 11 domains using PSIPRED software. The following sections describe the details of each subunit structural model of apoB-100, followed by information on its domain boundaries, the dynamics of its structure, and finally, the new functional insights provided by the model.

### 2.1. ApoB-100 Subunit I Homology Modeling

Human apoB-100 sequence (residues 28–1030) was applied to the HHpred server within the MPI bioinformatics toolkit for homology detection and structure prediction [26,27,28]. Templates with higher sequence homology probability (>95%) were selected and applied to HHpred-TemplateSelection to make a multiple sequence alignment in PIR format (Table 1) [26,27]. The generated alignment was forwarded to MODELLER for structural calculation [29]. Lipovitellin and microsomal triglyceride transfer protein (MTP) are two essential templates covering residues 43–1017 of apoB with a 100% sequence homology probability. I-TASSER server was used to fill the gaps in the N-terminus (residues 28–42) [30,31]. The model was created without intervention or manipulation by the experimental data, and the DSSO cross-link data and cysteines disulfide bonds were only used to validate the computational model. Considering the spacer length of DSSO 10.1 Å and the distance contribution by two lysine side chains 12.58 Å (2 × 6.29 Å) and 3.0 Å for backbone flexibility and structural dynamics; the Cα–Cα distances between DSSO cross-linked lysine residues is estimated to be ~26 Å [32,33]. The model was also evaluated for the seven known disulfide bonds located in subunit I. The sulfur groups of cysteine residues were correctly paired in the model and within the 5.6 Å constraint. Furthermore, 18 cross-links detected within this subunit measured within the 26 Å distance limit (Figure 1). Next, the structure was energy minimized to resolve any significant distortions using UCSF Chimera [34,35].

Subunit I computational model contains 990 amino acids (E28-S1017) in three distinct domains (Figure 1A). It resembles a pyramidal funnel with X:80 Å, Y:110 Å, Z:60 Å dimensions. Subunit I consist of an incomplete β-barrel at the top, two antiparallel β-sheets lined on two sides of the pyramid, supported by two curved layers of α-helices on the outer side. This model is similar to previously proposed models [16,18]. Subunit I domain1 (SID1, 28–320 aa.) consists of 11 β-strands that create an incomplete β-barrel with two α-helices on the side and within the center of the barrel (Figure 2A). The last 30 residues of SID1 make a random coil and connects the barrel to the second domain. Subunit I domain 2 (SID2, 321–616 aa.) consists of 17 amphipathic α-helices (Figure 2B). The helix structure is arranged in a double layer outside the β-sheets to apparently stabilize the core structure. Domain 2 mostly covers the right side, posterior outlet, and part of the left side of the pyramid (Figure 1A). Subunit I domain 3 (SID3, 617–1017 aa.) mostly forms the two sides of the pyramid and the base (Figure 2C). On the right side, residues 617–651 form a coil that connects the last α-helix from domain 2 to the first β-strand in domain 3. Five antiparallel β-strands make the β-sheet on the right side. A long coil extends from resides 693–772 to connect strand three to four. After the fifth strand, residues 790–820, made from two short helices and coils, connect the right and left β-sheets and cover the pyramid’s base. Eleven antiparallel β-strands make up the left side of the pyramid (Figure 1A).

Subunit I was evaluated by local cross-link data and known disulfide bonds in apoB by UCSF Chimera software [35,36]. For the 18 local cross-links that were identified by mass spectrometry, all Cα−Cα distances of DSSO cross-links between lysine residues were within 26 Å maximum distance constraint (Figure 1D). Moreover, for the seven known disulfide bonds in subunit I, all the cysteines were in close proximity (<5.6 Å) to form disulfide bonds before energy minimization. After optimizing the model, disulfide bonds formed among the paired cysteines (Figure 2D–F).

### 2.2. Subunit II Modeling

Human ApoB-100 sequence (residues 1018–2072) was applied to the I-TASSER server for structure prediction. Templates of the highest significance in the threading alignments include (PDB ID/chain): 4rm6A, 4acqA, 4u4jA, 7abiE, 2pffB, 1vt4A, 1kmiZ, 5n8pA, and 6ysaA [30,31]. Again, the model was generated without using the DSSO cross-link data. The model was evaluated for the 13 cross-links found in subunit II, and all the cross-links were found within the 26 Å distance limit (Figure 3A–C). Then, the structure was energy minimized by UCSF chimera to find and solve any steric clashes [34,35].

Subunit II model consists of 1055 residues (A1018-P2072) (Figure 3A). Subunit II coil (A1018-T1500) contains 483 residues and is almost entirely made of coils. This part can form various shapes in modeling and does not appear to have a preferred fixed structure (Figure 3A). Subunit II domain 1 (Y1501-P2072) is a boomerang shape β-helix with two wings. It is 572 residues long and is made from β-strands, turns, and coils. Domain 1 larger wing consists of 36 β-strands which make 16 turns with 90 Å length. The smaller wing is connected to the larger part with a ~90° angle. It is made from 17 β-strands that make eight turns with 60 Å length (Figure 3D). Domain 1 coronal section is oval in most parts, except for a few triangular areas, with 26 Å and 12 Å dimensions (Figure 3E). Sequence analysis of subunit II domain 1 by MPI Bioinformatics Toolkit HHrepID shows that this segment is made from 15 repeats (~28 residues each) and covers Q1528 to H1944 (*p*-value: 3.8 × 10^−14^) (Appendix A) [26,27,37].

### 2.3. Subunit III Homology Modeling

Human apoB-100 sequences (residues 2073–2273) and (residues 2274–2550) were applied to the HHpred server for homology detection [26,27,28]. Templates with >95% sequence homology probability were selected to make multiple sequence alignments in PIR format (Table 2). Generated alignments were forwarded to MODELLER for structural prediction [29]. Apolipoprotein E, apolipoprotein A-I, and apolipophorin-III proteins were the main templates used for modeling. Both models were generated without experimental data intervention. The structures energy minimized for steric clashes using UCSF Chimera. Then subunit III was assembled by coupling the two domains using DEMO within the I-TASSER server [38]. Top templates used for domain assembly include (PDB ID/chain): 3s5hA, 6lykA, 4av3A, 6hqvA, 3s5mA, 1iduA, 4bedA, 4bedB, 3uw8A, and 2ya0A. There are five intra-subunit cross-links. Three cross-links were found within the 26 Å limit, but two cross-links between K2100-K2387 and K2100-K2402 exceeded the 26 Å limit, 36 Å, and 34 Å, respectively (Figure 4C) [34,35].

Subunit III model consists of 478 residues (F2073-I2550) and two domains homologous to exchangeable apolipoproteins (Figure 4A,B). Domain 1 (F2073-I2273) and domain 2 (D2293-I2550) were entirely made from helix-turn-helix and coils.

### 2.4. Subunit IV Modeling

Human apoB-100 sequence (K2591-K4055) was entered into the I-TASSER server for modeling. The first forty residues (S2551-I2590) were excluded due to the sequence length and computational memory limit. Top threading templates used by I-TASSER include (PDB ID/Chain): 7abhE, 3b39A, 4o9xA, 7okqA, 4kncA, 3ljyA, 6ar6A, 6tgbA, and 2nbiA [30,31]. The model was generated without the intervention of DSSO cross-link data. The model was evaluated for the ten cross-links found in subunit IV. Seven cross-links were found within the 26 Å distance limit (Figure 5B,C). Two cross-links were slightly above the expected limit (K3946-K3973: 26.8 Å, K2853-K2829: 28 Å) and one cross-link was ~34 Å (K2911-K2926: 34.2 Å) (Figure 5B,C). The model was examined for one disulfide bond located in subunit IV (C3194-C3324). The sulfur groups of cysteine residues were 16 Å apart. After two steps of atomic-level energy minimization (Modrefiner) within the I-TASSER server and UCSF Chimera, cysteine residues were correctly paired and within the 5.6 Å constraint (Figure 5D) [34,35,39].

Subunit IV model includes 1507 residues (S2551-N4057). It begins with 350 residues in coil form (S2551-F2900) (Figure 5A). The first forty and last two residues were excluded from analysis due to computational processing limits. After the coil, the protein sequence transforms into strands that make three doughnut shape β-propeller domains (Figure 5A). Each domain makes a toroidal shape β-propeller system with symmetrical seven blades (β-sheets) around the central axis. Each blade typically has four antiparallel β-strands arranged so that the fourth strand is close to the center and perpendicular to the first strand (Figure 5A). All three domains have a similar diameter, around 46 Å and 30 Å thickness (Figure 5D). Domain one is 412 residues and includes W2929 through D3340. Domain 2 is 338 residues and begins at K3380 to Y3717. Domain 3 structure with 407 residues is more complex and in close contact with domain 1. Domain 3 is made from three separate sequences; it starts with S2901 to S2928, extends from F3341 to Y3379, and finally is completed by residues S3718 to N4057 (Figure 5E). Overall domain topology is that domain 1 and domain 2 central axis are almost parallel, whereas domain 3 axis is perpendicular to domains 1 and 2 (Figure 5E,F). Sequence analysis of residues A2610-E4059 by MPI Bioinformatics toolkit HHrepID shows that this segment is made from three repeats with 100% probability, 547 residues length, and *p*-value: 3.3 × 10^−228^ (Appendix A) [26,27,37].

### 2.5. Subunit V Homology Modeling

Human apoB-100 sequences (residues 4016–4270) and (residues 4260–4563) were applied to the HHpred server for homology detection [26,27,28]. Apolipoproteins with >90% sequence homology probability were selected as templates (Table 3). Generated alignments were forwarded to MODELLER for structural prediction [29]. The main templates used for the modeling were apolipoprotein E, apolipoprotein A-I, apolipoprotein A-IV, and apolipophorin-III. Both models were generated without experimental data intervention (Figure 6A,B). One intra-domain cross-link was found in the models, and it was within the 26 Å limit. The structure energy minimized for steric clashes using UCSF Chimera. Generated primary model was used as a template in I-TASSER to make the subunit V entire model (resides 4058–4563). Templates of the highest significance in the threading alignments include (PDB ID/chain): 4uxvA, 4iggA, 6thkA, 6z6fA, 6d03E, 7qj0H, and 1st6A [30,31]. The model was evaluated for the five cross-links found in subunit V, four cross-links were found within the 26 Å distance limit, and one cross-link was slightly over the limit (28 Å) (Figure 6E). Then, the structure was energy minimized by a two-step atomic-level energy minimization (Modrefiner) within the I-TASSER server and UCSF chimera to find and solve steric clashes [34,35,39].

Subunit V model includes 506 residues (W4058-L4563) with two domains (Figure 6). Subunit V domain 1 (W4058-Y4269) is made from eight amphipathic α-helices. Subunit V domain 2 with a similar structure composed of nine amphipathic α-helices partially perpendicular to domain 1.

### 2.6. Domain Boundaries Prediction and Secondary Structure Determination

In the early stages, subunits and domains boundaries were first defined by using DomPred within PSIPRED Server V4.0 [24,25]. After several steps examining secondary and tertiary structures, the boundaries between subunits and domains were determined. The results of apoB-100 subunit boundaries and secondary structure are summarized in Table 4. Subunit I, with 990 residues (21.8%), has an almost equal percentage of α-helix, β-strand, and coil. Subunit II, with 1055 residues (23.2%), begins with 483 amino acids, almost entirely in coil form. Domain 1 of this subunit is made from β-strands that create an oval shape β-helix. We considered the entire β-helix as β-strands. Subunit III, with 478 residues (10.5%), contains two domains, homologous to exchangeable apolipoproteins. Both domains lack β-strands and contain a similar secondary structure of α-helix and coil. Subunit IV with 1507 residues (33.2%) contributes the longest part of apoB-100. Like subunit II, it starts with a long coil, 350 aa., then creates three 7-bladed β-propeller systems that mostly contain β-strands and turns. Most of these domains were considered β-strands except for two short α-helices at the end of domain one and coils at the beginning, end, and between the domains. Domain V with 506 residues (11.2%) contains two domains with α-helix secondary structure homologous to exchangeable apolipoproteins. In summary, ~24% of the entire structure is α-helix, 41% β-strand, and 35% coil. These results sometimes differ from previous secondary structural studies. An LDL study by infrared spectroscopy at 37 ℃ estimated 24% α-helix, 23% β-sheet, 24% β-strand, 6% β-turns, and 24% unordered structure of human apoB-100 [40]. On the other hand, the circular dichroic spectrum of LDL study showed the helical content of apoB-100 is 25% to 33% [41]. In another evaluation of apoB-100 secondary structure in LDL by infrared spectroscopy, it was reported that the amount of β-sheet is 41% [42], identical to our findings.

### 2.7. Sequence and Structural Comparison to Lipovitellin

Lipovitellin is the predominant egg yolk lipoprotein in oviparous species. Electron microscopy reveals lipovitellin particles with a 4–6 nm diameter form a ring with an average size of 25 nm [43]. Studies show that lipovitellin evolutionary and structurally, is closely related to MTP and apoB-100. Furthermore, homology models for the apoB-100 amino-terminus 1000 residues based on available crystal structure have been made [16,17]. In this study, we made a computational model of silver lamprey’s lipovitellin and vitellogenin sequence, lipovitellin precursor, to fill up some of the crystal structure gaps and compare the lipovitellin crystal and computational model with our apoB-100 model. Lipovitellin structure data and vitellogenin sequence were acquired from the protein data bank and Uniprot and I-TASSER server was used to make the computational models (Appendix A) [30,31,43]. The first 1000 residues of the amino-terminus of lipovitellin and ApoB secondary and tertiary structure are very similar and resemble a pyramid (Appendix A). There are three breaks in this part of the lipovitellin crystal structure with no electron density (R708-W729, M759-A777, M949-F990) [44]. The computational model of lipovitellin shows that all three missing segments were coils (Appendix A). Furthermore, part of the lipovitellin that covers the base of the pyramid from K1306-H1355 with no electron density is also a coil within the computational model [44]. In addition, segments of the vitellogenin sequence that were missing in the lipovitellin crystal structure, including a serine-rich area of vitellogenin (P1074-F1305), due to proteolytically cleaved protein or lack of electron density in the crystal structure were entirely coil in the computational model (Appendix A) [44,45,46,47]. According to the apoB computational model, the secondary structure of residues A1018-T1500 is a coil and comparable to vitellogenin sequence P10740-H1355 (Appendix A). Moreover, lipovitellin β-strands which form the β-sheet and cover the base of the lipid cavity (S1356-F1529) are comparable to apoB subunit II domain 1 secondary structure (Y1501-P2072) (Appendix A).

### 2.8. ApoB-100 Architecture and Dynamics

The length of apoB-100 made it impossible to orient all subunits in one step using currently available bioinformatic tools. Therefore, different subunits and domains were extended and combined from C-terminus and N-terminus, with and without cross-link data, to optimize long-distance cross-link coverage. Then, large structures were examined and validated to see if they could be fitted manually. Making large segments reduced the accuracy of domains, especially within domains and subunits boundaries, but it was necessary for designing the whole architecture and validating subunits folding.

The first segment (E28-T1500) was generated by the I-TASSER server applying K763-K1324 and K1087-K1121 DSSO cross-links to orient the subunit II coil and estimate the coil location and dynamics (Figure 7A–C). The second segment covers Y1501-Y2256, generated by the same tool [30,31]. Then, the two components were oriented using cross-links C (K766-K2208) and E (K923-K1593) (Figure 7A–C). Both cross-links’ distances were <26 Å, and subunit II was positioned very close to subunit I, and they did not clash. Since the two segments were generated independently without using the cross-link data (except for the coil part 1018–1500) and all cross-links were within the 26 Å limit, we concluded that the structural match geometrically was a strong proof that both models were accurate enough to fit by applying the cross-link data. Furthermore, we generated different protein folding for the second segment to compare with the later model and see if those models could fit with segment one using cross-links C and E. The models were generated without applying cross-link data by I-TASSER and RoseTTAFold. Even though both models’ intra-domain cross-links were within the 26 Å limit, neither of those models fit well with segment one using cross-links C and E (Appendix A) [30,31,48]. The third (Q2257-T2548) and fifth (W4058-L4563) segments were positioned next by using cross-links I (K2100-K2387 and K2100-K2402), A, and B (K1696-K4349 and K1958-K4207). The structures did not clash, and all cross-links’ distances were <26 Å (Figure 7D). The fourth segment was added using cross-links G and H (K2270-K3210 and K4207-3148 and K4207-3159). The cross-links were <26 Å, and all five segments did not clash (Figure 7E). Cross-link F, a coil on one side (K2671-K1586 and K2671-K1593), was adjusted by changing residues torsion in the coil region (K2591-F2900). Cross-link J (K3682-K4103) was not within the 26 Å limit (102 Å) (Figure 7E). It is possible that the subunit IV domain 2 model is the mirror image of the actual structure. Therefore, it was manually repositioned on the other side of subunit IV domain 3 to fill the gap between the two subunits, and cross-link J became <26 Å (Figure 7F). Twelve out of thirteen inter-subunit cross-links were within the 26 Å limit in the final model, and cross-link D, which connects subunit II coil to subunit I was 37 Å. A total number of 65 unique cross-links were identified by mass spectrometry. The distance between Cα−Cα of lysine residues was measured for 64 cross-links. Fifty-six (87.5%) cross-links were within the 26 Å threshold, three cross-links were slightly beyond the 26 Å limit (26.9 Å, 2 × 28.0 Å), and for five cross-links Cα−Cα of lysine residues was >30Å (33.9 Å, 34.2 Å, 36.3 Å, 37.3 Å, 42.5 Å). For the last two cross-links, at least one lysine residue was located within the coil region (Appendix A).

ApoB-100 model was arranged on the LDL particle according to the cross-link data from Figure 7 and published CryoEM data (Figure 8A,B) [21]. The LDL lipid is represented as a discoidal shape (20 nm × 20 nm × 11 nm) with a round shape from the top (Figure 8A) and an oval shape from the side views (Figure 8B). Segments one (E28-T1500) and two (Y1501-Y2256) were arranged on the top of the disc. Subunit II coil (A1018-T1500), which is part of segment one, is mostly located below subunit I and connects the latter part to subunit II domain 1 (Y1501-P2072) (Figure 8A). Segment three (Q2257-T2548) is on the front side, segment four (K2591-K4055) is on the bottom and right side, and segment five (W4058-L4563) is on the back side of the disk (Figure 8A,B). Subunit IV coil (K2591-F2900) that connects subunit III to subunit IV domain 3 is located on the right and bottom sides. There is a gap of 42 residues (Y2549-I2590) which is part of the subunit IV coil. These residues were excluded from analysis because of the software processing limitation from the subunit IV long sequence. Cryo-EM studies show that the LDL receptor β-propeller domain binds to the linker part of apoB-100 on the right side of the LDL particle, which connects the top and bottom parts of the protein [21]. This finding suggests the β-propeller domain of the LDL receptor binds to a β-propeller domain of subunit IV located on the right side of the LDL particle (Figure 8A,B).

ApoB-100 model on VLDL was also designed according to the immunoelectron microscopy ribbon and bow model and DSSO cross-link data (Figure 8C,D) [49]. This model is more hypothetical compared to the LDL model due to less data. The VLDL lipid is represented in a spherical shape with a 30 nm diameter. Segments 3 and 5 with α-helical structures are more extended than globular, and the subunit II coil is unrolled instead of folded in the LDL model (Figure 8). The rest of the apoB-100 structure that was applied for the VLDL model was the same as the LDL model. In the proposed model, segments 1–4 (E28-K4055) form an incomplete ring around the spherical lipid particle, and segment 5 (W4058-L4563) moves in the opposite direction and crosses segment 4. The VLDL proposed model is almost entirely consistent with the immunoelectron microscopy results. Furthermore, the immunoelectron microscopy model supports subunit II domain 1 folding with the boomerang or V shape. This part starts around R1507 (apoB-32) and ends around M2358. There is a “kink” in the middle of this part at M1881. According to the article, the protein starts at apoB-2, then encircles the spherical shape lipid, then at apoB-41 (Kink), it significantly changes its direction through apoB-50 [49]. Interestingly, the “kink” region perfectly matches the apoB model and the kink start point (M1881) is where the two wings of boomerang shape subunit II joins (Figure 8C).

A number of studies mentioned that apoB-100 architecture is likely very dynamic and flexible. An important potential factor that causes apoB conformational changes is the surface pressure alteration due to size change from lipolysis [18,50]. Lipoprotein particle interface surface pressure progressively increases during the transformation of VLDL to LDL [50]. Thus, it was suggested that α-helix rich subunits (III and V) are expanded and in contact with the surface lipids in VLDL. While during VLDL conversion to LDL, the pressure rises, and α-helix subunits come off the surface to possibly form a globular conformation. Unlike α-helix subunits, β-strand rich subunits (II and IV) are predicted to be tightly anchored to the core lipid and keep the protein bound to the lipid [18,50]. Furthermore, we propose that coils in apoB-100, especially subunit II and subunit IV coils, play a key role in the protein flexibility and architecture adjustment due to size and surface pressure changes. Since in a large size lipoprotein, there is less surface pressure on apoB-100, we hypothesized that five subunits are expanded over the lipid surface and far apart from each other. Subunit II coil with 483 residues and subunit IV coil with 351 residues are similar to an open rope on the opposite sides. The two long coils connect subunit I to subunit II and subunit III to subunit IV, respectively (Figure 8C). We suggest that during VLDL to LDL conversion, coils on both sides are remodeled from open to a folded form and bring the five subunits to close proximity (Figure 8A–D). This structure alteration regulates and reduces the pressure on the entire structure to maintain its integrity.

According to DSSO cross-link data, VLDL lacks the entire distal cross-links and many short-distance cross-links except for the N-terminus β-barrel (Appendix A). VLDL and LDL DSSO cross-link data comparison support the computational models of apoB-100 architectures on LDL and VLDL. Moreover, the disappearance of short-distance cross-links in VLDL might be due to submerged protein into VLDL lipids compared to smaller LDL particles. In addition, LDL negative staining electron microscopy micrographs show LDL particles with various morphologies such as circular, oval, and rectangular. On the other hand, VLDL particles are reported to be more circular than rectangular [21,23].

In another attempt to determine the protein’s hydrophilic sites, hydrolyzed DSSO cross-links and painted apoB-100 sequences in LDL particles were analyzed. The hydrolyzed DSSO cross-links were almost entirely located in the painted regions, which indicates the hydrophilic sites by applying two different methods. Both methods show that the protein hydrophilic and hydrophobic patches are not uniform and lack specific orientation (Appendix A).

### 2.9. β-propeller Folds and Apob-100 Docking Region

β-propeller structures are a type of all beta-strand proteins made of four to twelve β-sheets called blades. Depending on the blade number, they have a wide range of functions. For instance, β-propellers with six and seven blades, in addition to structural and ligand binding functions, can operate as a signaling protein, lyase, hydrolase, or as oxidoreductase enzyme [51,52,53,54,55,56,57]. Interestingly, several reports mentioned the possible role of lipoproteins in various non-traditional activities, such as modulating signal transduction and exocytosis, DNA binding and transfection capacity, lyase activity, and oxidoreductase activity [3,5,6,58,59]. Moreover, subunit IV seven-bladed β-propellers are complementary to the six-bladed β-propeller domain of LDL-receptor YWTD-EGF (Figure 9) [60]. Additionally, as was mentioned above, according to the Cryo-EM studies and the theoretical LDL model, the LDL receptor β-propeller domain binds to a β-propeller domain of subunit IV (Figure 8A,B) [21]. According to a study, monoclonal antibodies specific for apoB-100 epitopes between P2953-K3057, near residue A3473, and between S4000-I4054 are inaccessible on receptor-bound LDL. According to the proposed computational model, those regions are located on subunit IV domains 1–3, respectively. Conversely, a monoclonal antibody specific for apoB-100 between F2808-I2895 could bind the epitope on receptor-bond LDL, in accordance with the proposed model part of subunit IV coil just before β-propellers start point, S2901. ApoB-100 docking architecture supports the idea that the binding site is multivalent and that the domains can act in concert or independently (Figure 5) [61]. In addition, the apoB-100 β-propeller domain proposed model can also create a pocket for calcium ions in the center, which helps explain the mechanism of LDL binding to LDL-receptor [62].

## 3. Methods and Materials

### 3.1. Sequences and Domain Prediction

The human apolipoprotein B100 reference sequence (P04114) was acquired from the Uniprot database [63]. Protein templates used in the study are available at the Protein Databank. Domain boundaries were predicted by applying DomPred (Protein Domain Prediction) and PSIPRED Server V4.0 by searching against the database with 0.01 PSI-BLAST e-value cutoff and five iteration [24,25]. Since the human ApoB-100 sequence is 4563 residues long, it is not feasible to generate the model in one step with the current software and computer processing limitations. On the other hand, breaking down the protein sequence into many small pieces can significantly affect the models’ accuracy and make it difficult to dock the parts together. Therefore, several strategies have been applied to determine the domain boundaries by software prediction and examining secondary and tertiary structures in every step. Since the maximum sequence length that PSIPRED can process is 1500 residues, the entire sequence was divided into three fragments of 1500 residues length each, starting from the first amino acid (1–1500, 1501–3000, 3001–4500). In addition, the entire sequence was examined for domain boundaries in segments of 500 residues length starting from the first residue and sliding to the next possible domain boundary. For instance, the first segment began at 1–500, and the second segment started from 321–820 by knowing that position 320 is a potential domain boundary. Then the results acquired by each method were compared with each other through the entire modeling process as well as their secondary and tertiary structure to optimize the results.

### 3.2. Database Search and Modeling

Human ApoB 100 structure is a heterogeneous molecule. Therefore, searching for templates and modeling for different parts varies based on the available structures in the database. The most successful protein structure prediction method relies on identifying homologous templates with known structures [26,27]. However, a database search does not provide homologous templates for all the regions. In this case, servers based on fold recognition by threading were used to generate models. Different software, such as the MPI Bioinformatic Toolkit, I-TASSER, DEMO, RoseTTAFold, and Phyre2 web portals, were used to develop accurate models [26,27,38,48,64,65,66]. Homologous template database searches and selections are critical points for optimal alignment; hence, different parameters, including probability, E-value, and identity, have been modified for each sequence search to extend the target sequence coverage without reducing accuracy. Several models were generated from the sequence alignment with homologous templates. Then each model’s secondary and tertiary structures were examined and validated by experimental data, such as disulfide bonds and DSSO crosslinks. Finally, the high-ranked models for all domains were selected and assembled (Figure 10).

### 3.3. Plasma Lipoprotein Separation by Sequential Ultracentrifugation

To fractionate lipoproteins (VLDL, LDL, HDL) by ultracentrifugation, a single batch of fresh whole plasma from a healthy donor was provided by the NIH blood bank. Lipoproteins were isolated from fresh plasma sample by sequential potassium bromide density ultracentrifugation according to the procedure [67]. Various lipoproteins were collected carefully after every ultracentrifugation step. Harvested samples were dialyzed with 10K MW cassettes in PBS buffer at 4 °C to remove potassium bromide. Collected samples were stored at 4 °C and used for cross-linking.

### 3.4. Cross-Linking Assay

Disuccinimidyl sulfoxide (DSSO), a mass-spec cleavable cross-linker, was used to test and validate individual computational models. As described above, VLDL and LDL subfractions were obtained by density gradient ultracentrifugation from a healthy donor. Dimethyl sulfoxide (DMSO) and 20mM HEPES 4-(2-hydroxyethyl)-1-piperazineethanesulfonic acid) (pH 7.5) were added to samples to optimize the reaction according to the manufacturer’s protocol. DSSO (SA246004, ThermoFisher Scientific^®^, Rockford, IL, USA) was prepared just before adding samples by dissolving a 1 mg vial with 50 µL dimethyl sulfoxide (DMSO). Samples were incubated at room temperature for one hour. The reaction was quenched with 2 µL 1M/Tris buffer for 15 min at room temperature to remove the extra free crosslinker, and then stored at 4 °C. DSSO was added to samples accordingly and incubated at room temperature for one hour. The product results were run on an SDS-PAGE 4–12% to validate cross-linked molecules.

### 3.5. Sample Preparation for Mass Spectrometry with Double Trypsin Digestion

Protein concentration in samples was measured with a NanoDrop 1000 spectrophotometer (Thermo Fisher Scientific, Waltham, MA, USA). An amount of 50 µg of total protein (100 µL) from each cross-linked sample was applied for further processing. Trypsin (V5111, Promega, Madison, WI, USA) was added to each sample with a protease: protein ratio of 1:50 (*w*/*w*) and incubated at 37 °C for one hour. Partially digested samples were transferred to glass tubes for delipidation with a chloroform: methanol ratio of 2:1 (*v*/*v*). Samples were incubated on ice for 30 min. After adding cold methanol, they were centrifuged at 4000× *g* for 20 min at 4 °C. Supernatants were discarded, and pellets were dissolved in cold methanol and centrifuged as above. Delipidated protein precipitates were resuspended in (100 µL) 8M urea and diluted by 50 mM NH4HCO3 (PH 7.8) to reach 1 M urea concentration. Samples were reduced by adding 200 mM (40 µL) dithiothreitol (DTT) and incubated at 37 °C for 30 min. Then carbamidomethylated by 800 mM (40 µL) iodoacetamide (IA) and incubated for 30 min at room temperature in dark. For a second digestion step, trypsin was added in a protease: protein ratio of 1:50 (*w*/*w*), digest overnight at 37 °C. Digestion reaction was terminated by adding formic acid. Samples were concentrated by Speed-Vac until the final volume was 100 µL. Peptides were desalted and purified using C18 resin, ZipTip^®^ (ZTC18S096, Millipore Sigma, Burlington, MA, USA) according to the manufacturer’s protocol, dried in Speed-Vac, and stored at −20 °C. Samples were reconstituted in 20 µL 0.1% formic acid and transferred into sample vials (C1411-13, Thermo Fisher Scientific, Waltham, MA, USA) for mass spectrometry.

### 3.6. Mass Spectrometry

Desalted tryptic peptides were analyzed using nanoscale liquid chromatography tandem mass spectrometry (nLC- MS/MS) and Ultimate 3000-nLC online coupled with an Orbitrap Lumos Tribrid mass spectrometer (Thermo Fisher Scientific, Waltham, MA, USA). Peptides were separated on an EASY-Spray C18 column (Thermo Fisher Scientific, Waltham, MA, USA, 75 μm by 50 cm inner diameter, 2-μm particle size, and 100-A°pore size). Separation was achieved by 6 to 35% linear gradient of acetonitrile +0.1% formic acid for 60 min. An electrospray voltage of 1.9 kV was applied to the eluent via the EASY-Spray column electrode. Eluting peptides were analyzed by a Tribid Fusion Lumos mass spectrometer (Thermo Fisher Scientific), using a hybrid CID-MS/MS, and CID-MS3 fragmentation scheme, as previously described (1). Briefly, MS scans were acquired in data dependent mode on the Orbitrap (resolution = 120,000, maximum injection time = 50 ms, AGC target = 4.0 × 10^5^, scan range (*m*/*z*) = 375–1500). The duty cycle was restricted to 5 s. For each selected MS1 precursor (z = 4–8, dynamic exclusion = 30 s, intensity ≥ 2.5 × 10^−4^), CID (normalized collision energy of 25%) fragmentation was performed in sequential scans. MS/MS scans were recorded at an Orbitrap resolution of 30,000. A mass difference (DSSO reporter ion doublets) of 31.79 amu detected in CIDMS/MS was used to trigger up to 4 CID-MS3 rapid ion-trap analyses (normalized collision energy 35%).

### 3.7. Mass Spectrometry Data Analysis

The .raw file was analyzed using Proteome Discoverer 2.3 with XlinkX^1^ v2.0 nodes and searched against an in-house curated LDL human protein database (60 proteins known to associate with LDL). Xlink algorithm identifies MS-cleavable crosslinkers and the detection settings were as follows: Acquisition strategy: MS2_MS3, Crosslink Modification: DSSO/+ 158.004 Da (K), Minimum S/N: 1.5. Xlink Search settings were: Max. Missed Cleavages: 3, Max.

Peptides Considered: 1, Min. Peptide Length: 5, Min. Peptide Mass: 300, Max. Peptide Mass: 7000, Min. score: 40 unless otherwise stated, Min. Score Difference = 4 unless otherwise stated, Precursor Mass Tolerance: 12 ppm, FTMS Fragment Mass Tolerance: 20 ppm, FTMS.

Fragment Mass Tolerance: 0.6 Da, Allowed Precursor Mass Offset: 0, Allowed Precursor.

Signal to Noise: 10, Static Modification Carbamidomethyl +57.021 Da (C), Dynamic.

Modification: Oxidation +15.995 Da (M). Xlink Validator settings were: FDR threshold: 0.02, FDR Strategy: Simple, Search concatenated: False. In the Consensus, XlinkX Crosslink Grouping node settings were: Minimum score threshold: 20, Ignore reporter scan identification: False. Combinations of minimum score and minimum score difference settings in the XlinkX Search node were used to generate datasets of different stringencies. MS2 CID MS/MS spectra were also searched for linear peptides against the LDL human protein database using the Sequest HT node within Proteome Discoverer 2.3. Relevant settings included: Max. missed cleavages 3, Min. peptide length,= 6, Max. peptide length = 150, Max. number of peptides considered = 10, precursor mass tolerance = 10 ppm, Fragment mass tolerance = 0.02 Da, Dynamic modification = Oxidation (M) +15.995 Da, DSSO Hydrolyzed (K) +176.014 Da. Static modification = Carbamidomethyl I +57.021 Da. Target Decoy was used for PSM FDR validation, using a concatenated target/decoy strategy, validation based on q-value and a strict FDR of 1%, as well as the Protein FDR Validator node used for protein FDR validation. Confident intra-protein crosslinks for APOB crosslinked spectral matches (CSMs) were considered [68].

### 3.8. Protein Painting

Painted and unpainted ApoB samples were prepared in triplicate from a single batch of LDL particles isolated from a healthy donor. Protein concentration was 1.7 mg/mL. LDL particles (10 µg per sample) were pulsed with 50 µL of a 1 mg/mL solution of freshly prepared covalent paint molecule Fast Blue B (Cat# D9805, Sigma Aldrich, St. Louis, MO, USA) for 5 min. Unpainted control samples were pulsed with 50 µL of PBS. Following painting, samples were passed over a pre-washed Sephadex G-25 gel filtration column (Cat# 11814397001, Sigma Aldrich, St. Louis, MO, USA) according to the manufacturer’s directions. The gel-filtered samples were added to a glass test tube containing 500 µL of 2:1 chloroform: methanol and incubated on ice for 30 min. Following incubation, an additional 400 µL of ice-cold methanol was added, samples were mixed via vigorous pipetting, then transferred to a 2 mL Eppendorf microcentrifuge tube and centrifuged at 4000× *g* for 20 min at 4 °C. Supernatant was removed and the pellet was washed and resuspended in 1 mL ice-cold methanol, then centrifuged at 4000× *g* for 20 min 4 °C. The resulting protein pellet was resuspended in 75 µL of 8M urea. Samples were subsequently reduced with 10 mM DTT, alkylated with 50 mM iodoacetamide, and digested with trypsin at a 1:10 protease: protein ratio for 1.5 h. Digested peptides were desalted using Pierce C18 Columns (Cat# 89873, ThermoFisher Scientific, Waltham, MA, USA) according to the manufacturer’s directions. Samples were analyzed via mass spectrometry as previously reported (1). Briefly, LC-MS/MS experiments were performed on an Orbitrap Fusion (Thermo Fisher Scientific, Waltham, MA, USA) equipped with a nanospray EASY-LC 1200 HPLC system (Thermo Fisher Scientific, Waltham, MA, USA). Peptides were separated using a reversed-phase PepMap RSLC 75-μm inner diameter × 15 cm long with 2 μm, C18 resin LC column (Thermo Fisher Scientific, Waltham, MA, USA). The mobile phase consisted of 0.1% aqueous formic acid (mobile phase A) and 0.1% formic acid in 80% acetonitrile (mobile phase B). After sample injection, the peptides were eluted by using a linear gradient from 5 to 50% B over 30 min and ramping to 100% B for an additional 2 min. The flow rate was set at 300 nl/min. The mass spectrometer was operated in a data-dependent mode in which one full MS scan (60,000 resolving power) from 300 Da to 1500 Da using quadrupole isolation was followed by MS/MS scans in which the most abundant molecular ions were dynamically selected by Top Speed and fragmented by collision-induced dissociation using a normalized collision energy of 35%. “Peptide Monoisotopic Precursor Selection” and “Dynamic Exclusion” (8 s duration), were enabled, as was the charge state dependence so that only peptide precursors with charge states from +2 to +4 were selected and fragmented by collision-induced dissociation. Tandem mass spectra were searched using Proteome Discover version 2.1 with SEQUEST against the NCBI *Escherichia coli* and *Saccharomyces cerevisiae* databases and a custom database containing sequences of the recombinant proteins using tryptic cleavage constraints. Mass tolerance for precursor ions was 5 ppm, and mass tolerance for fragment ions was 0.06 Da. Data were analyzed with oxidation (+15.9949 Da) on methionine as a variable post-translation modification and carbamidomethyl cysteine (+57.0215) as a fixed modification. A 1% false discovery rate was used as a cut-off value for reporting peptide spectrum matches (PSMs) from the database. Regions of paint coverage were identified as peptides present in all three unpainted samples and absent in all three painted samples [69].

## 4. Conclusions

We developed a “divide and conquer” algorithm using PSIPRED software by dividing apoB into five subunits and 11 domains. Models of each domain were prepared using different structural prediction software. Then, experimentally, 64 DSSO cross-links and eight disulfide bonds were used to validate each model. A total of 56 cross-links (87.5%) were within the 26 Å threshold, and all eight known disulfide bonds were paired and within the expected 5.6 Å constraint. Finally, multiple domains and subunits were combined to merge smaller domains. We also examined the dynamics of apoB-100 as it transitions from VLDL to LDL. As first suggested by J. Segrest [70], our data also indicates a significant folding difference in some regions of lipovitellin compared to apoB. In the proposed model, the base of subunit I is covered by the first 1000 residues and residues 1500–2072 fold in a β-helix form instead of a simple β-sheet like lipovitellin. A significant factor for the change could be lipid size differences in LDL and VLDL (18–95 nm) compared to lipovitellin (4–6 nm). Therefore, two β-sheets may have evolved to form a β-helix with enough strength and flexibility to stabilize the larger lipoprotein particle. We also showed that subunit IV is made from three repeats in the form of seven-bladed β-propeller domains, which may enable apoB-100 to bind to various sites. According to the suggested model, ~24% of the entire structure is α-helix, 41% β-strand, and 35% coil. The two long coils in subunit II and subunit IV may account for the high flexibility of apoB-100. In addition, the computational model of subunit II domain 1 simple β-sheet did not match with the DSSO cross-link data (Appendix A). Furthermore, the “kink” region in the immunoelectron microscopy ribbon and bow model perfectly matched the subunit II domain 1 model [49]. Studies have shown that LDL and apoB-100 containing lipoproteins not only constitute a significant atherosclerosis risk factor but also induce the signaling in different tissues, including vascular smooth muscle and lung alveolar cells [3,5,6]. Additionally, the similarity between lipoproteins and hepatitis C virus structure raises the possibility that this molecular mimicry of HCV may play an essential role in its life cycle [4]. Finally, this study represents the first use of the “divide and conquer” computational algorithm integrated with cross-linking. This approach may be valuable for analyzing other complex and large proteins.

## Figures and Tables

**Figure 1 ijms-23-11480-f001:**
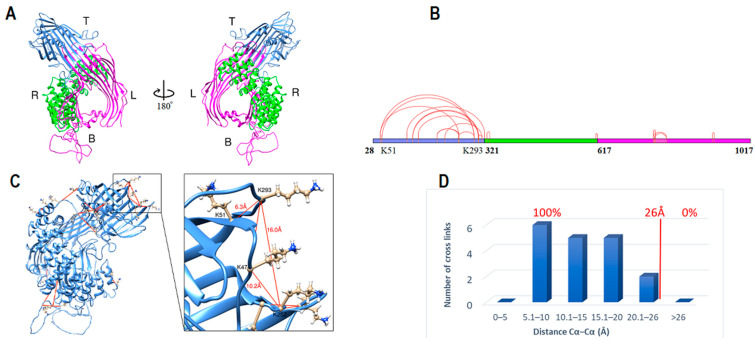
ApoB-100 subunit-I structure and cross-link map. (**A**) Ribbon representation of ApoB-100 subunit I structure. The anterior view of subunit I followed by 180° rotation (posterior view). Domain 1 forms an incomplete β-barrel at the top (blue). Domain 2 with an α-helices on the right (green) and domain 3 forms the left side and the base (magenta). (**B**) DSSO cross-link map on subunit I linear sequence with three domains is colored according to A. (**C**) DSSO cross-link map on subunit I ribbon representation with 18 intra-subunit cross-links (red lines) between lysine residues. (**D**) The distribution plot of identified DSSO cross-links in subunit I vs. the spatial distances of lysine residues in the computational model.

**Figure 2 ijms-23-11480-f002:**
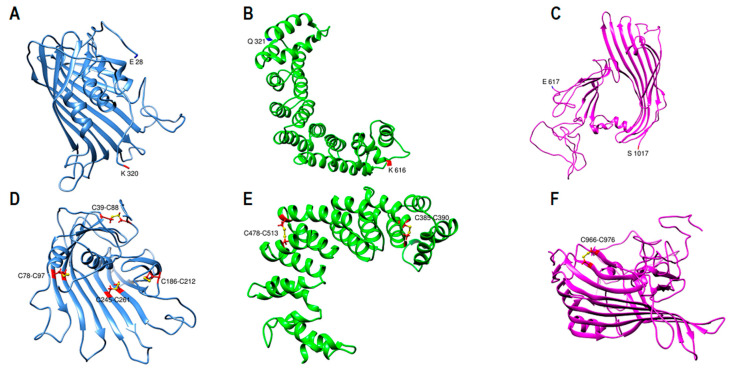
ApoB-100 subunit I domains structures and disulfide bonds. (**A**–**C**) Ribbon representation of subunit I domains. The N and C termini of each domain are colored in dark blue and red respectively and labeled with a residue number. Domain 1 (blue) is an incomplete β-barrel that extends from E28 residue to K320. Domain 2 (green) is arranged in two layers of α-helices (Q321-K616). Domain 3 (magenta) with two β-sheets on the right and left side and α-helices at the base creates a cavity in the center. (**D**–**F**) Ribbon representation of subunit I domains with disulfide bonds colored in yellow and cysteine residues colored in red with the ball and stick representation. Four disulfide bonds are located in domain 1 (**D**), two disulfide bonds in domain 2 (**E**), and one disulfide bond in domain 3 (**F**).

**Figure 3 ijms-23-11480-f003:**
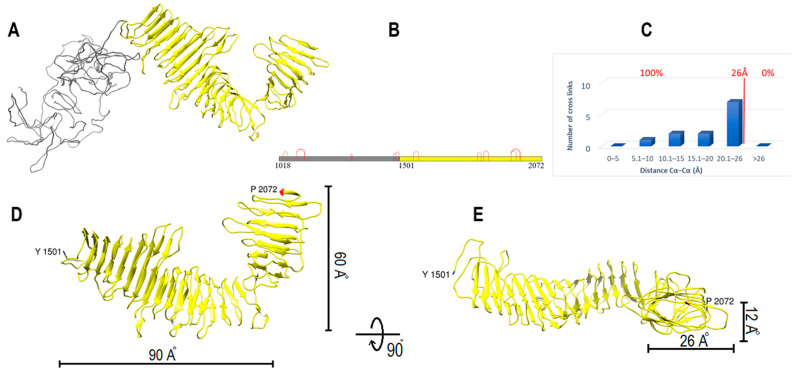
ApoB-100 subunit II structure and cross-link map. (**A**) Ribbon representation of ApoB-100 subunit II coil (gray) and domain 1 (yellow) with β-helix folding. (**B**) DSSO cross-link map on subunit II linear sequence colored according to A. (**C**) The distribution plot of identified DSSO cross-links in subunit II vs. the spatial distances of lysine residues in the computational model. (**D**,**E**) Ribbon representation of subunit II domain 1 shows a boomerang shape β-helix with 90 Å and 60 Å wings span followed by a 90° rotation representation. Coronal section of this domain shows an oval shape helix with 26 Å and 12 Å dimensions.

**Figure 4 ijms-23-11480-f004:**
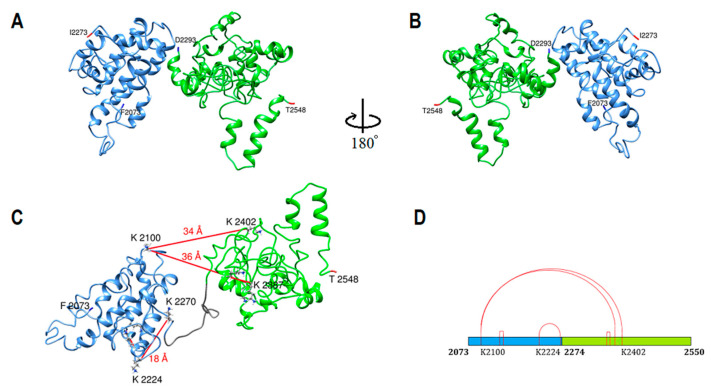
ApoB-100 subunit III structure and cross-link map. (**A**,**B**) Ribbon representation of ApoB-100 subunit III structure. Domain 1 (blue) begins at F2073 and extends to I2273. The shorter form of domain 2 (green) begins at D2293 through T2548. The entire structure is composed of helix-turn-helix motifs. Both domains were generated by MODELLER using apolipoproteins as templates. (**C**) DSSO cross-link on subunit III ribbon representation with five intra-subunit cross-links (red lines) between lysine residues. Two cross-links between K2100-K2387 and K2100-K2402 exceeded the expected 26 Å limit. (**D**) DSSO cross-link map on subunit III linear sequence with two domains is colored according to C without considering the coil sequence that connects domain 1 to domain 2.

**Figure 5 ijms-23-11480-f005:**
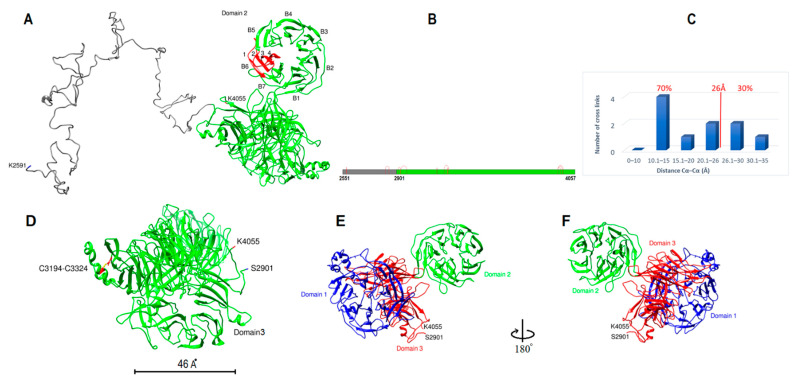
ApoB-100 subunit IV structure and cross-link map. (**A**) Ribbon representation of ApoB-100 subunit IV (gray) with coil secondary structure and β-propeller domains (green). Domain 2 seven β-sheet blades are shown (B1–B7) around the central axis. Antiparallel β-strands of blade 6 (red) are shown that the central strand (4th strand) is perpendicular to the first strand. (**B**) DSSO cross-link map on subunit IV linear sequence with two domains is colored according to A. (**C**) The distribution plot of identified DSSO cross-links in subunit IV vs. the spatial distances of lysine residues in the computational model. Seven out of ten (70%) cross-links were found within the 26 Å limit, two cross-links were slightly above the limit, and one cross-link was ~34 Å. (**D**–**F**) Ribbon representation of subunit IV domains shows the architecture of three domains. (**D**) shows the location of disulfide bond within domain 1 between C3194-C3324. The diameter of each domain is ~46 Å. (**E**,**F**) Domains 1 and 3 are predicted to be in close contact, and domains 1 and 2 central axis are parallel and perpendicular to domain 3 axis.

**Figure 6 ijms-23-11480-f006:**
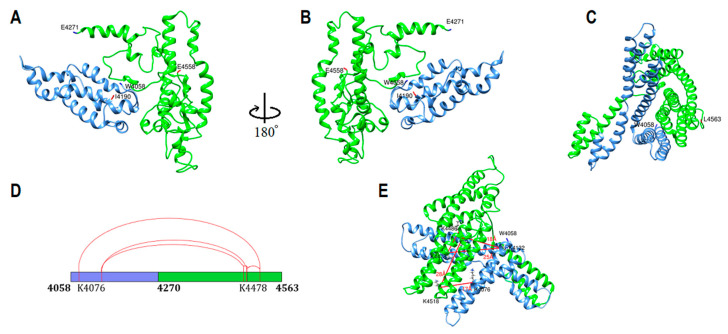
ApoB-100 subunit V structure and cross-link map. (**A**,**B**) Ribbon representation of ApoB-100 subunit V primary structure generated by homology modeling. Domain 1 (blue) starts at W4058 through I4190, and domain 2 (green) begins at E4271 through E4558. (**C**) Entire subunit V generated by I-TASSER using the primary structure as a template. Subunit V structure is composed of helix-turn-helix and short coils. (**D**) DSSO cross-link map on subunit V linear sequence with two domains is colored according to the structure. (**E**) DSSO cross-link map on subunit V ribbon representation with five intra-subunit cross-links (red lines) between lysine residues. One cross-link between K4485-K4518 has slightly exceeded the 26 Å limit (28 Å).

**Figure 7 ijms-23-11480-f007:**
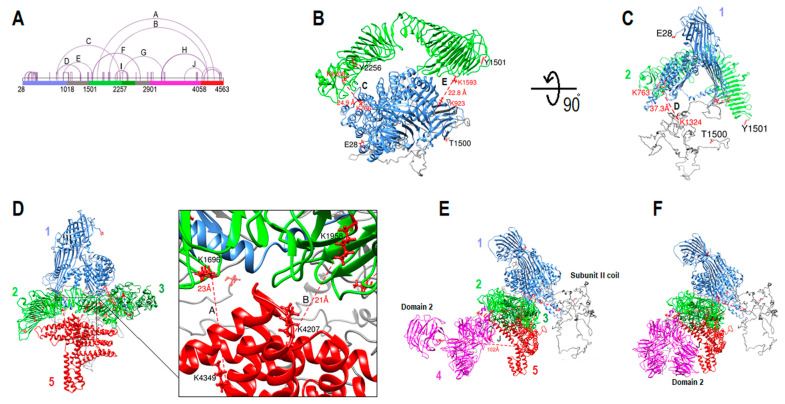
ApoB-100 architecture and cross-link map. (**A**) DSSO cross-link map of linear apoB-100 sequence with different regions colored according to the structure segments. Inter-subunit cross-links are labeled from A–J. (**B**) Cross-links C and E were used to arrange segment one (E28-T1500) and segment two (Y1501-Y2256). (**C**) Cross-link D was applied in I-TASSER to estimate subunit II coil location. Colorful numbers from 1–5 close to the structure indicate different segments. (**D**) Cross-links I, A, and B were used to position segments three (Q2257-T2548) and five (W4058-L4563). (**E**) Segment four (K2591-K4055) was the last part of locating by applying cross-links G, and H. Cross-link F that connects subunit II domain 1 to subunit IV coil was arranged by coil sequences torsion adjustment (not shown). Cross-link J (K3682-K4103), which connects subunit IV domain 2 to subunit V was the only inter-subunit linker that did not fit within the 26 Å limit (102 Å). (**F**) Subunit IV domain 2 was manually repositioned on the other side of subunit IV domain 3 to fill the gap between subunits IV and V. After repositioning domain 2 cross-link J fit within the 26 Å limit (25 Å).

**Figure 8 ijms-23-11480-f008:**
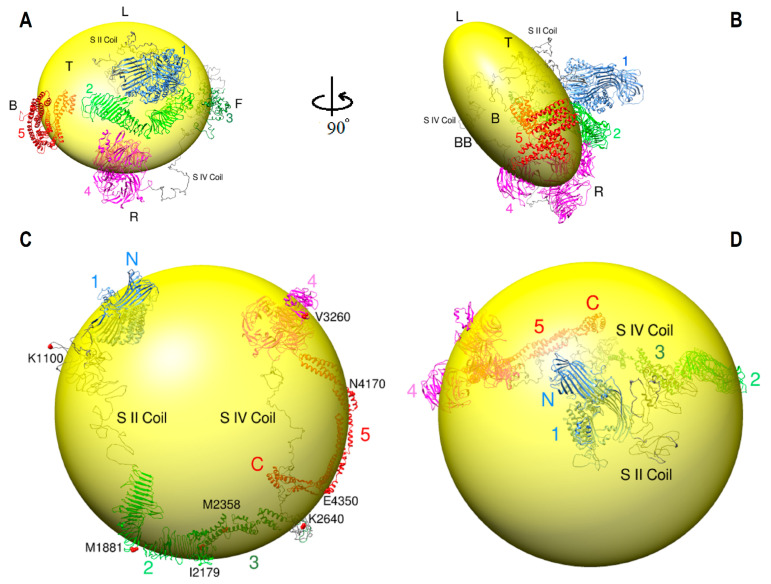
ApoB-100 architecture on lipoprotein particles. (**A**,**B**) Ribbon representation of apoB-100 on discoidal LDL (20 nm × 20 nm × 11 nm). The LDL particle is round from the top (Figure A) and oval shape from the side views (Figure B). Segments one (1) and two (2) are located on the top (T) of the disc. Subunit II coil (S II coil) is folded and positioned beneath subunit I. Segment three (3) on the front (F), segment four (4) on the bottom (BB) and right (R), and segment five (5) are located on the back (B). The protein does not cover the LDL particle’s left (L) side. (**C**,**D**) Ribbon representation of apoB-100 on spherical VLDL with 30 nm diameter. The structure encircles the lipid particle starting from the N-terminus (N) through segment four (4), then moves in the opposite direction (5) and crosses segment four and ends at the C-terminus (C). The red spherical representation of residue M1881 indicates the kink start point through M2358. Part of the protein from the N-terminus to I2179 illustrates apoB-48.

**Figure 9 ijms-23-11480-f009:**
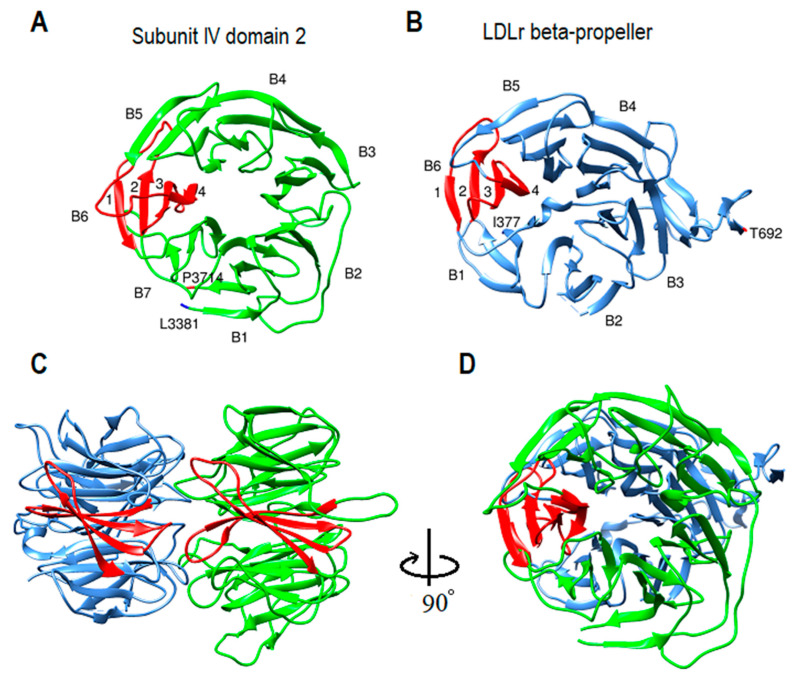
ApoB-100 subunit IV domain 2 comparison with low-density lipoprotein receptor (LDLr) β-propeller domain. (**A**) Ribbon representation of the computational model of the toroidal shape subunit IV domain 2 β-propeller begins at L3381 through P3714. This domain comprises seven blades (B1–B7), and each blade is made from four β-strands. (**B**) Ribbon representation of the crystal structure of the LDLr β-propeller domain begins at I377 through T692 and consists of six blades (B1–B6) [60]. (**C**,**D**) The side and top aspects of the apoB-100 domain and LDLr domain show that both domains have a similar structure and are complimentary.

**Figure 10 ijms-23-11480-f010:**
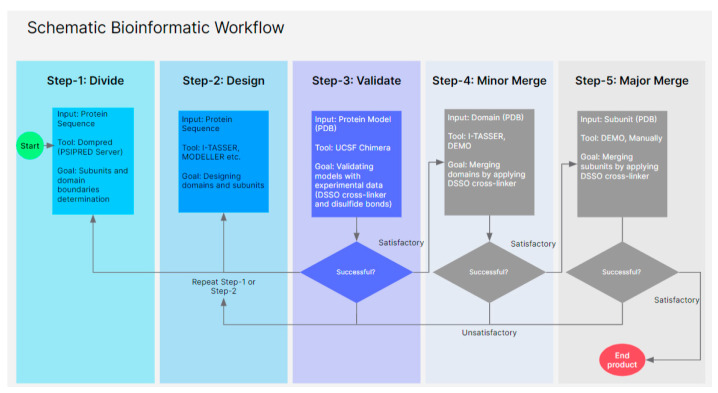
Brief depiction of a “divide and conquer” workflow applied to apoB-100 and related structures.

**Table 1 ijms-23-11480-t001:** Selected homologous templates for Subunit I.

PDB ID/Chain	Molecule	Classification	Probability	E-Value	*p*-Value
1LSH_A	Lipovitellin	Lipid binding protein	100.0	4 × 10^−107^	5 × 10^−112^
6I7S_G	Microsomal triglyceride transfer protein	Lipid transport	100.0	1 × 10^−73^	1.3 × 10^−78^
4D50_A	Deoxyhypusine hydroxylase	Oxidoreductase	98.0	0.0062	8.1 × 10^−8^
5N3U_A	Phycocyanobilin lyase	Lyase	97.2	0.11	1.4 × 10^−6^
4XL5_C	bGFP-A	Protein binding	97.0	0.56	7.3 × 10^−6^
6QH5_A	AP-2 complex subunit alpha	Protein transport	97.0	0.33	4.2 × 10^−6^
4L7M_B	Putative uncharacterized protein	Unknown function	96.9	0.38	5 × 10^−6^
5NZR_K	Coatomer subunit gamma-1	Transport protein	96.9	0.54	7 × 10^−6^
5FUR_I	Transcription initiation factor TFIID	Transcription	96.8	0.057	7.4 × 10^−7^
6QH5_B	AP-2 complex subunit beta	Protein transport	96.8	0.92	1.2 × 10^−5^
2DB0_B	253aa long hypothetical protein	Protein binding	96.7	0.62	8 × 10^−6^
6YAF_B	AP-2 complex subunit beta	Endocytosis	96.7	0.41	5.3 × 10^−6^
6GWC_C	IE5 ALPHAREP	Cell cycle	96.7	0.81	1 × 10^−5^
1JDH_A	Beta-catenin	Transcription	96.4	0.87	1.1 × 10^−5^
5XJG_A	Vacuolar protein 8	Signaling protein	96.3	0.48	6.3 × 10^−6^
5N3U_B	Phycocyanobilin lyase subunit beta	Lyase	96.3	0.27	3.6 × 10^−6^
1OYZ_A	Hypothetical protein yibA	Structural genomics	95.4	0.69	9 × 10^−^^6^
1TE4_A	Conserved protein MTH187	Structural genomics	95.3	0.72	9.3 × 10^−6^

Human apoB-100 sequence (residues 28–1030) was entered into HHpred (https://toolkit.tuebingen.mpg.de/tools/hhpred) (accessed on 30 April 2021) for homology search. Templates with probability >95% were selected for multiple sequence alignment using HHpred-TemplateSelection.

**Table 2 ijms-23-11480-t002:** Selected homologous templates for subunit III model.

PDB ID/Chain	Molecule	Classification	Probability	E-Value	*p*-Value
2L7B_A	Apolipoprotein E	Lipid transport	96.2	1.1	2 × 10^−5^
1LS4_A	Apolipophorin III	Lipid transport	96.0	0.98	1.7 × 10^−5^
3R2P_A	Apolipoprotein A-I	Lipid transport	95.9	0.91	1.6 × 10^−5^
3R2P_A	Apolipoprotein A-I	Lipid transport	96.8	0.37	6.4 × 10^−6^
2L7B_A	Apolipoprotein E	Lipid transport	96.4	1.00	1.8 × 10^−5^
5VJ4_A	Uncharacterized protein	Lipid binding protein	96.4	0.58	1 × 10^−5^
2LEM_A	Apolipoprotein A-I	Lipid transport	96.2	0.99	1.7 × 10^−5^

Human apoB-100 (residues 2073–2273) and (residues 2274–2550) were separately entered to HHpred for homology detection. The first three templates were selected for subunit III domain 1 and the last four for subunit III domain 2 modeling.

**Table 3 ijms-23-11480-t003:** Selected homologous templates for subunit V primary model.

PDB ID/Chain	Molecule	Classification	Probability	E-Value	*p*-Value
1EQ1_A	Apolipophorin III	Lipid binding protein	93.44	4	7 × 10^−5^
1KMI_Z	Chemotaxis protein	Signaling protein	91.98	6.7	1.2 × 10^−4^
2L7B_A	Apolipoprotein E	Lipid transport	97.6	0.14	2.5 × 10^−6^
3S84_A	Apolipoprotein A-IV	Transport protein	97.3	0.27	4.7 × 10^−6^
2LEM_A	Apolipoprotein A-I	Lipid transport	97.1	0.24	4.1 × 10^−6^
3K2S_B	Apolipoprotein A-I	Lipid binding protein	96.9	0.64	1.1 × 10^−5^
3R2P_A	Apolipoprotein A-I	Lipid transport	96.6	0.72	1.3 × 10^−5^

Human apoB-100 (residues 4016–4270) and (residues 4260–4563) were separately entered to HHpred for homology detection. The first two templates were selected for subunit V domain 1 and the last five templates for subunit V domain 2 modeling.

**Table 4 ijms-23-11480-t004:** Subunits prediction and secondary structure content.

Subunit/Domain	Residue Number	α-Helix (Percentage of Total Content)	β-Strand (Percentage of Total Content)	Coil (Percentage of Total Content)
Subunit I	28–1017	7%	6.5%	~8%
Subunit II	1018–2072	<0.5%	12.5%	10.5%
Subunit III	2073–2550	~5.0%	0%	~5.0%
Subunit IV	2551–4057	~1%	22%	10.5%
Subunit V	4058–4563	~10.5%	0%	~1%
Total	28–4563	~24%	41%	35%

Human apoB-100 subunits and secondary structure prediction. The secondary structure was calculated from final models using UCSF Chimera.

## Data Availability

We will freely provide data to all qualified investigators.

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
