# Peer review of "A New Structural Model of Apolipoprotein B100 Based on Computational Modeling and Cross Linking"

_ijms, 2022, doi:10.3390/ijms231911480_

Round 1

Reviewer 1 Report

I have found this paper very interesting and intriguing.  ApoB-100 is the largest of all the apolipoproteins (4563). Nowdays there are no AlphaFold model or other relevant models for the whole structure. Modelling a structure for this protein looks a very hard task. I think the authors choosen appropriate methods for the modelling. Cross-linking assay is a very strong argument for modelling results.   The article is well written, clear and logical. I like the paper, but have some comments and questions:

1. Please, provide descriptions for VLDL and LDL abbreviations.

2. Please, provide the link for downloading the whole structure file. I cannot view and estimate the quality of the structure without file. 

Author Response

Dear Reviewer,

The authors would like to thank you for your comments. We have addressed each question point-by-point in our submitted response.

  1. Please, provide description for VLDL and LDL abbreviations.

The authors described abbreviations in the revised version of the manuscript.

  1. Please, provide the link for downloading the whole structure file. I cannot view and estimate the quality of the structure without file.

We have uploaded seven PDB files of the structure within the Zipped supplementary data available to download. Five PDB files for five subunits with high resolution and accuracy, one file for the extended subunit-II 1018-2256, and one file for the entire apoB-100 28-4563. We should mention that these are the essential files we used to assemble the whole structure. The apoB file that covers the entire structure (28-4563) was used for designing figure 8C-D, and it is more based on theoretical modeling than experimental data. Instead, the five subunit files are more based on experimental data. As we mentioned on page 11, lines 353-359, “ApoB-100 large molecule made it impossible to orient all subunits in one step using currently available bioinformatic tools. Therefore, different subunits and domains were extended and combined from C-terminus and N-terminus, with and without cross-link data, to optimize long-distance cross-link coverage. Then, large structures were examined and validated to see if they could be fitted manually. Making large segments reduced the accuracy of domains, especially within domains and subunits boundaries, but it was essential to design the whole architecture and validate subunits folding.” Since we used several files to design the model, please let us know if you want to evaluate any specific file from the manuscript.

Sincerely,

Reviewer 2 Report

The authors proposed a divide and conquer-based novel method for determining a new structural model of Apolipoprotein B100. 

The manuscript contains the following issues:
 1. Authors should include a schematic diagram of the methodological pipeline so that it becomes easier to follow the flow of the method

 2. Authors should mention the PDB IDs of the templates used.

 3. Authors should justify whether or not the proposed method is heavily dependent on the structure prediction software used.

 4. References have formatting issues. After 63, it started again from 1!! 

Author Response

Dear Reviewer,

We thank you for your insightful suggestions, which have resulted in an improvement of our article. We have addressed each question point-by-point in our submitted response.

  1. Authors should include a schematic diagram of the methodological pipeline so that it becomes easier to follow the flow of the method.

We have designed a Schematic Bioinformatic Workflow and included it in the method section as figure 10 on page 16.

  1. Authors should mention the PDB IDs of the templates used.

Here we addressed the PDB IDs in the manuscript as below:

  • Page3 Table1 “Table 1. Selected homologous templates for Subunit I.” PDB IDs of 18 templates are listed.
  • Page5 line165 “(PDB ID/chain): 4rm6A, 4acqA, 4u4jA, 7abiE, 2pffB, 1vt4A, 1kmiZ, 5n8pA, 6ysaA”. List of top templates used by I-TASSER for designing subunit II.
  • Page6 lines200-201 “(PDBID/chain): 3s5hA, 6lykA, 4av3A, 6hqvA, 3s5mA, 1iduA, 4bedA, 4bedB, 3uw8A, 2ya0A”. List of top templates used by DEMO for designing subunit III.
  • Page6-7 Table2 “Table 2. Selected homologous templates for subunit III model.”. These 7 templates were selected to design subunit III domain 1 and subunit III domain 2 primary models.
  • Page7 line224 “(PDB ID/Chain): 7abhE, 3b39A, 4o9xA, 7okqA, 4kncA, 3ljyA, 6ar6A, 6tgbA, 2nbiA”. List of top templates used by I-TASSER for designing subunit IV.
  • Page9 line275-276 “(PDB ID/chain): 4uxvA,4iggA, 6thkA, 6z6fA, 6d03E, 7qj0H, 1st6A”. Top templates used by I-TASSER for designing subunit V.
  • Page9 Table3 “Table 3. Selected homologous templates for subunit V primary model.”. These 7 templates were selected to design subunit V domain 1 and subunit V domain 2 primary models.

  1. Authors should justify whether or not the proposed method is heavily dependent on the structure prediction software used.

Yes, the proposed method heavily depends on the software that was used. Since various modeling software has different algorithms, the software results are not the same. Sometimes, the proposed models for the same sequence made by two modeling tools have completely different secondary and tertiary structures. Because of apoB-100 long sequence, heterogeneous structure, and computer processing limitations, we decided to split the protein from domain boundaries. And as we mentioned in the method section on page 16, line 538, “The most successful protein structure prediction method relies on identifying homologous templates with known structures. However, database search does not provide homologous templates for all the regions. In this case, servers based on fold recognition by threading were used to generate models. Different software such as MPI Bioinformatic Toolkit, I-TASSER, DEMO, RoseTTAFold, and Phyre2 web portals were used to develop accurate models”. Furthermore, modeling tools vary in strength and weakness, which is unknown in many cases. Therefore, we had to try different software and validate the results with experimental data. Then after comparing different models, choose the most accurate ones.

  1. References have formatting issues. After 63, it started again from 1!!

This technical issue was due to downloading the method section as a separate file. We were able to correct the references in the revised manuscript.

Sincerely,
